# The Sub-Nuclear Localization of RNA-Binding Proteins in KSHV-Infected Cells

**DOI:** 10.3390/cells9091958

**Published:** 2020-08-25

**Authors:** Ella Alkalay, Chen Gam Ze Letova Refael, Irit Shoval, Noa Kinor, Ronit Sarid, Yaron Shav-Tal

**Affiliations:** The Mina & Everard Goodman Faculty of Life Sciences and The Institute of Nanotechnology and Advanced Materials, Bar-Ilan University, Ramat Gan 5290002, Israel; alkalayella@gmail.com (E.A.); gamzolc@gmail.com (C.G.Z.L.R.); irit.shoval@biu.ac.il (I.S.); noit.kinor@gmail.com (N.K.); Ronit.Sarid@biu.ac.il (R.S.)

**Keywords:** KHSV, nuclear speckles, SRSF2, nuclear structure, RNA

## Abstract

RNA-binding proteins, particularly splicing factors, localize to sub-nuclear domains termed nuclear speckles. During certain viral infections, as the nucleus fills up with replicating virus compartments, host cell chromatin distribution changes, ending up condensed at the nuclear periphery. In this study we wished to determine the fate of nucleoplasmic RNA-binding proteins and nuclear speckles during the lytic cycle of the Kaposi’s sarcoma associated herpesvirus (KSHV). We found that nuclear speckles became fewer and dramatically larger, localizing at the nuclear periphery, adjacent to the marginalized chromatin. Enlarged nuclear speckles contained splicing factors, whereas other proteins were nucleoplasmically dispersed. Polyadenylated RNA, typically found in nuclear speckles under regular conditions, was also found in foci separated from nuclear speckles in infected cells. Poly(A) foci did not contain lncRNAs known to colocalize with nuclear speckles but contained the poly(A)-binding protein PABPN1. Examination of the localization of spliced viral RNAs revealed that some spliced transcripts could be detected within the nuclear speckles. Since splicing is required for the maturation of certain KSHV transcripts, we suggest that the infected cell does not dismantle nuclear speckles but rearranges their components at the nuclear periphery to possibly serve in splicing and transport of viral RNAs into the cytoplasm.

## 1. Introduction

Kaposi’s sarcoma-associated herpesvirus (KSHV), also known as human herpesvirus 8 (HHV-8), is a member of the lymphotropic gammaherpesvirus subfamily and one of seven known oncogenic human viruses [1]. This obligatory parasite is the etiologic agent of Kaposi’s sarcoma (KS), an angioproliferative multifocal neoplasm of the skin and other organs, that predominantly occurs in immunodeficient individuals, such as AIDS patients, but also among apparently healthy individuals, in particular the elderly. KSHV is also causally associated with two lymphoproliferative diseases, namely primary effusion lymphoma (PEL) and multicentric Castleman’s disease (MCD), as well as with a condition that mimics acute severe sepsis termed KSHV-inflammatory cytokine syndrome (KICS) [2,3,4,5,6].

KSHV can potentially infect a wide variety of human cell types such as keratinocytes, epithelial, endothelial and lymphoid cells, as well as several types of animal cells [7]. Like all herpesviruses, replication and transcription of the KSHV double-stranded DNA genome take place in the nucleus of the infected cells [8,9]. Once the viral genome reaches the host nucleus, one of two alternative infection cycles may occur: latent or lytic. During latent infection the viral DNA exists in the cell nucleus in the form of a circular chromatinized episome, no new virions are produced and only a small proportion of the KSHV gene repertoire is expressed. Latent gene products assist in maintaining the viral genome by tethering it to the host chromosomes, supporting its replication and directing its segregation into daughter cells during mitosis, thus establishing a long term infection [10,11,12]. In contrast, the lytic infection cycle involves extensive viral DNA replication which ends by the assembly and release of new virions. This cycle is characterized by temporal expression of most KSHV genes, which are classified as immediate-early (IE), early (E) and late (L) genes based on their transcription kinetics. Lytic infection results in dramatic reorganization of the nuclear architecture involving marginalization of host chromatin [13,14], which enables the formation of nuclear viral replication compartments where viral DNA replication and capsid assembly take place. In addition, as the virus hijacks the host machineries to preferentially produce viral transcripts, transcription and splicing factors are found adjacent to these replication compartments [15].

The nucleus of mammalian cells contains a variety of nuclear bodies that harbor various RNA-binding proteins [16,17]. Immunostaining with antibodies to splicing factors in cells under regular conditions has shown that splicing factors are diffusely dispersed in the nucleoplasm and are enriched in sub-nuclear bodies termed nuclear speckles [18,19,20,21,22,23,24]. These dynamic structures are located throughout the nucleus of mammalian cells and their number may be over 20, serving not only as a hub for splicing factors but also for kinases and a variety of other nuclear factors involved in mRNA biogenesis. The exact function of these nuclear bodies is not known, but spatially they can be closely associated with active genes [25,26,27]. RNA fluorescence in situ hybridization (FISH) performed with probes that hybridize to the polyA tail of RNAs has shown that nuclear speckles are enriched in polyadenylated RNAs [28,29], of which some are long non-coding RNAs (lncRNAs) such as MALAT1 in nuclear speckles and Neat1 in paraspeckles [30]. Nuclear speckles have been suggested to function as storage and recycling centers for splicing factors returning from splicing to be re-phosphorylated by kinases [31,32]. Recently, it was suggested that nuclear speckles may regulate the levels of splicing factors in the nucleoplasm and thereby control gene activity by modulating the availability of splicing factors in the nucleoplasm [24,33].

Pre-mRNA splicing is a vital process occurring in all higher eukaryotes that increases transcriptome diversity and complexity of the single organism. The core splicing factors are the U1, U2, U4, U5 and U6 snRNPs that interact directly with the pre-mRNA [34]. Many other auxiliary splicing factors impact spliceosome function, and some of these belong to the family of serine/arginine (SR)-rich proteins [35,36]. The SR family is comprised of 12 factors termed Serine Arginine Splicing Factors (SRSFs) that share two main features: an ‘RNA recognition motif’ (RRM) at the N-terminal of the proteins, which binds RNA, and a ‘Serine Arginine (SR) domain’ at the C-terminus, which can be phosphorylated, and promote interactions of SR proteins with RNA segments at active genes or with other proteins [37].

SR proteins facilitate the binding of the U1 and U2 snRNP spliceosome core complexes to the 3′ and 5′ splice sites in the pre-mRNA molecule [38,39] and contribute to the stabilization of the early spliceosomal complex. Furthermore, they take part in escorting the U4/U6 U5 snRNPs to the spliceosomal complex during the later stages of splicing [40]. SR proteins can also promote alternative splicing by binding to alternative exons [41]. Finally, since most SR proteins can bind multiple exons, they compete with each other such that their binding to the pre-mRNA can amplify or repress the inclusion of a certain exon, thus broadly affecting splicing [42]. An important SR protein is Serine Arginine Splicing Factor 2 (SRSF2) also known as SC35, which is used as a prominent marker of nuclear speckles. This splicing factor enables either constitutive splicing or alternative splicing for weak exon upon binding to regulatory sequences at the pre-mRNA, thereby determining transcripts fate [39]. Since the lytic cycle of KSHV infection is associated with modification of the global distribution of chromatin in the host nucleus, we were interested in examining the localization of RNA-binding proteins that function in gene expression pathway, in particular, during splicing and mRNA export. Furthermore, we aimed to determine the fate of sub-nuclear structures and their components during the lytic cycle of KSHV infection.

## 2. Materials and Methods

### 2.1. Cell Culture and Viruses

Human SLK or iSLK-PURO cells kindly provided by Prof. Don Ganem, (University of California) and Prof. Rolf Renne (University of Florida) [43] were maintained in Dulbecco’s modified Eagle’s medium (DMEM) supplemented with 10% fetal bovine serum (HyClone Laboratories, Logan, UT, USA), 50 IU/mL penicillin and 50 μg/mL streptomycin (Biological Industries, Kibbutz Beit Haemek, Israel). 

To maintain the Tet-On transactivator and K-Rta expression cassette in iSLK cells, 250 μg/mL G418 and 1 μg/mL puromycin (A.G. Scientific Incorporation, San Diego, CA, USA), respectively, were added to iSLK-PURO growth medium. Recombinant viruses BAC16 GFP (kindly provided by Prof. Jae J. Jung, University of Southern California) [44] and BAC16 GFP-mCherry-ORF45 were previously described [45]. Infected iSLK-PURO were selected with 600 μg/mL Hygromycin (Sigma-Aldrich, St. Louis, MI, USA).

Lytic reactivation was induced with 50 ng/mL Doxycycline (Dox) (Sigma-Aldrich) and 1 mM n-Butyrate (Sigma-Aldrich) for 48 or 72 h. Inhibition of viral DNA replication was achieved by adding 0.5 μM Phosphonoacetic acid (PAA) (Sigma-Aldrich) to the culture 2 h before lytic reactivation and during reactivation (48 h).

### 2.2. Immunofluorescence

5 × 10^5^ cells were grown on 18 mm coverslips (Bar Naor Ltd. Ramat-Gan, Israel) in 12-well plates. Cells were washed with 1xPBS (Invitrogen, Waltham, MA, USA) and fixed for 15 min in 4% paraformaldehyde (PFA) (Electron Microscopy Science, Hatfield, PA, USA). Alternatively, cells were fixed with ice cold methanol (5 min) (Daejung, Shieung-si, South Korea) and acetone (2 min) (Carlo Erba Reagents, Val de Reuil, France), for U1A, SF3B1 and Prp8 staining. After fixation, cells were permeabilized in 0.5% Triton X-100 ((Sigma-Aldrich) for 2 min and blocked with 0.5% bovine serum albumin (MP Biomedical, Solon, OH, USA). After blocking, cells were immunostained for 1 h or overnight with a primary antibody, and after subsequent washes with 1× PBS, the cells were incubated for 45 min with a secondary fluorescent antibody. 

Primary antibodies: Mouse anti-SC35 (S4045; 1:1000) (Santa Cruz, Dallas, TX, USA), mouse anti-Aly (A9979; 1:500) (Sigma), mouse anti-RNA Pol II (H14 hybridoma), rabbit anti-U1A (ab155054; 1:500), rabbit anti-U2AF65 (ab37530; 1:100), rabbit anti-SF3B1 (ab170854; 1:100, rabbit anti-SON (ab121759; 1:3500), rabbit anti-Prp8 (ab79237; 1:400), mouse anti-Y14 (ab5828; 1:200), rabbit anti-UAP56 (ab47955; 1:150), mouse anti-Nup62 (ab610497; 1:600), rabbit anti-Nup153 (ab84872; 1:300), rabbit anti-PABPN1 (ab75855 1:150) (Abcam, Cambridge, United Kingdom). 

Secondary antibodies: Goat anti-rabbit (ab6939), goat anti-mouse IgG H&L (Cy3) (ab97035) (Abcam), goat anti-rabbit IgG (H+L) cross-adsorbed AlexaFluor 594, goat anti-rabbit IgG (H+L) cross-adsorbed AlexaFluor 647, goat anti-mouse IgG (H+L), Superclonal™ recombinant AlexaFluor 647 (Invitrogen). Nuclei were counterstained with Hoechst 33342 (Sigma-Aldrich) and coverslips were mounted in mounting medium. In certain cases, immunofluorescence was performed after RNA FISH using the standard protocol.

### 2.3. Western Blotting

Plated cells were reactivated for a period of 24, 48 and 72 h. During harvesting, the cells were washed in cold phosphate-buffered saline (PBS), and suspended in RIPA lysis buffer containing the protease inhibitors PMSF and a commercial cocktail (Roche Diagnostics, Little Falls, NJ, USA). Cells in lysis buffer were incubated on ice for 20 min, and centrifuged at 11,000 × g for 15 min at 4 °C. Sodium-dodecyl sulfate (SDS) loading buffer was added, and the samples were boiled for 10 min. Protein lysates (30 μg) were resolved by SDS-PAGE and transferred to a nitrocellulose membrane (0.45 μm) using a transfer apparatus (Bio-Rad, Berkeley, CA, USA). The membrane was blocked with 5% BSA and then probed with a primary antibody for 2 h or 1 h at room temperature (RT), followed by incubation with HRP-conjugated goat anti-rabbit/mouse IgG (Sigma) for 1 h at RT. Immunoreactive bands were detected by the Enhanced Chemiluminescence kit (ECL, Pierce Waltham, MA, USA). Primary antibodies used: mouse anti-Y14 (Abcam) and rabbit anti-Tubulin (Abcam).

### 2.4. RNA Fluorescence In Situ Hybridization (FISH)

Cells were grown on coverslips, reactivated and fixed for 15 min in 4% PFA as described above. Seventy percent ethanol was added to the plates that were left at 4 °C overnight. The next day, cells were washed with 1× PBS and treated for 2 min with 0.5% Triton X-100. Cells were washed with 1× PBS and incubated for 10 min in 15% formamide (4% SSC) (Sigma-Aldrich). Cells were hybridized overnight at 37°C in 15% formamide with a specific fluorescently-labeled Oligo-dT Cy3-labeled DNA probe (~10 ng probe, 50 mer). The next day, cells were washed twice with 15% formamide for 15 min and then washed for two hours with 1× PBS. Nuclei were counterstained with Hoechst 33342 and coverslips were mounted in mounting medium. 

FISH experiments with Stellaris probes (Biosearch Technologies, Novato, CA, USA) for MALAT1 (labeled with Quasar^®^ 570 dye), Neat1 (Quasar^®^ 670), *orf29, orf50, orf57* (Quasar^®^ 670) were performed according to the manufacturer’s adherent cell protocol. *orf29* and *orf57* probes were targeted to the exons of the mRNAs, while ORF50 probes targeted the single *orf50* intron (958 bases). As iSLK-PURO cells contain an exogenous genomic insertion of *orf50* cDNA (lacking intron), which promotes lytic virus induction upon Doxycycline treatment, we designed probes to the intronic region of the *orf50* mRNA, which is expressed only upon lytic virus induction. To reduce photobleaching, the cells were submerged in GLOX buffer (pH = 8, 10 mM, 2x SSC, 0.4% glucose), supplemented with 3.7 ng of glucose oxidase (Sigma-Aldrich G2133-10KU) and 1 μl Catalase (Sigma-Aldrich 3515), prior to imaging.

### 2.5. Image Analysis and Statistical Analysis

Measuring the volume of SRSF2 nuclear speckles was performed in three dimensions (3D) using the IMARIS 9.5 software (Bitplane, Zurich, Switzerland), to identify foci with the “Surface” tool applied on each untreated/reactivated cell. SRSF2 foci (647 nm channel) that were adjacent were split into single foci. Colocalization of SRSF2 foci or PABPN1 with oligo-dT foci was performed using Imaris. First, both the SRSF2/PABPN1 foci (channel 647 nm) and oligo-dT foci (channel Cy3) were segmented by using the Imaris “3D surfaces” module. Once segmented, the extent of SRSF2/PABPN1 foci and oligo-dT foci co-localization was quantified using the “Surface-surface coloc” extension (3rd Party Xtension ‘MSD Analysis’ from Matthew J. Gastinger, Bitplane, Abingdon, UK) and compared between untreated and reactivated cells. For the counting of nuclear speckles, iSLK cells latently infected with a recombinant BAC16 mCherry-ORF45 KSHV clone were left untreated or treated for 48 h with n-Butyrate and Doxycycline to induce lytic reactivation. After immunostaining with the SRSF2 antibody, nuclear speckles were counted per nucleus of control and reactivated cells. In order to calculate Pearsons’ r colocalization coefficients, the ImageJ JaCoP plugin was used [46]. For each of the proteins tested, identical regions of interest (ROIs) in each channel were cropped in the images. The “Pearsons’ coefficient” option was checked. Van Steensel’s CCF and X shift of 20 was used as a control.

The experiments in this study were repeated at least three times. For statistical analysis of data in the plots, independent sample T-test (two-tailed) were preformed using the SPSS software on cells from each treatment. A Levene’s test to assess the equality of variances determined that equal variances cannot be assumed (*p* < 0.05) and the two tails test showed *p* < 0.001.

## 3. Results

### 3.1. Nuclear Speckles Aggregate at Condensed Chromatin Upon Lytic Reactivation of KSHV

Lytic KSHV infection has been previously shown to be associated with changes in the chromatin and nuclear structures of the infected host cell [15,47]. We aimed to determine the distribution of key RNA binding proteins (RBPs), which are often found in the nucleoplasm and in nuclear speckles known to contain many types of splicing factors, during the lytic KSHV infection cycle. KSHV BAC16-infected iSLK cells that mostly carry latent infection were treated with Doxycycline (Dox) and n-Butyrate for 72 h to induce lytic reactivation. In this setting, Doxycycline induces the expression of ORF50, which functions as a viral replication and transcription activator controlled by TetOn/rtTA (reverse tetracycline-controlled transactivator), while n-Butyrate further enhances lytic reactivation by inhibiting histone deacetylases [43]. Indeed, changes in DNA architecture were observed, involving the condensation of chromatin and its marginalization at the nuclear periphery (Figure 1A). Next, nuclear speckles were stained with an antibody to the SR splicing factor SRSF2 (SC35), an established marker of these structures. Over 20 irregularly shaped nuclear speckles were detected in untreated cells, whereas a dramatic reduction in their numbers, reaching up to one large nuclear speckle per nucleus, was observed in nuclei of cells undergoing lytic viral infection cycle (Figure 1B and Appendix A). The enlarged nuclear speckles showed a rounded shape, and often, some were very large, likely formed by the fusion of several nuclear speckles. Under regular conditions, nuclear speckles are typically found in the nuclear interior, but in nuclei undergoing lytic infection, some of the nuclear speckles re-localized to the periphery of the nucleus adjacent to the condensed chromatin. Changes in the structure of the nuclear speckles were only observed in cells presenting chromatin marginalization that characterizes advanced stages of lytic reactivation [15,47]. To rule out any effects of Doxycycline and n-Butyrate on nuclear speckle organization, we similarly treated un-infected parental SLK cells for 24, 48 or 72 h with these compounds. As expected, there was no difference between untreated iSLK cells and treated control uninfected cells (Appendix A).

To further verify whether the changes in nuclear speckle organization were dependent on lytic reactivation of KSHV, we used a KSHV recombinant clone that encodes the lytic viral protein ORF45 fused to monomeric Cherry fluorescent protein (mCherry), and thus enables tracking of cells undergoing lytic infection. Indeed, enlarged SRSF2 nuclear speckles appeared only in cells where mCherry-ORF45 was expressed and the chromatin was condensed (Appendix A). 

In order to determine whether these changes in nuclear speckle structure and distribution could be observed at earlier time points after lytic induction, we examined cells after 48 h after treatment. It is important to note that lytic reactivation is an asynchronous event. Accordingly, treatment with Doxycycline and n-Butyrate induces in certain cells lytic reactivation soon after treatment, while other cells enter the lytic cycle later and others do not reactivate at all. Therefore, 48 h after treatment a collection of cells that are either latent or undergoing lytic reactivation at different stages is observed. We focused on cells containing condensed chromatin, which marks late stages of lytic reactivation, and obtained similar results (Appendix A). Therefore, further experiments were conducted within this time frame. The numbers and volumes of the nuclear speckle foci were counted and measured in control untreated and induced BAC16-mCherry-ORF45-infected iSLK cells. As shown if Figure 1C, the numbers of SRSF2-stained nuclear speckles in reactivated cells was significantly lower than in control cells and dropped from an average of 25 nuclear speckles per nucleus (with a minimum of 15 per nucleus) to less than 6 in cells undergoing lytic reactivation (Figure 1C). Concurrently, the volume of the enlarged nuclear speckles in reactivated cells peaked to a maximum of 27 µm^3^ compared to 4 µm^3^ in control untreated cells (Figure 1D).

### 3.2. Changes in the Distribution of Splicing Factors during Lytic Reactivation of KSHV Infection

Other than SRSF2, nuclear speckles hubs are enriched with factors belonging to the splicing machinery, namely snRNPs and SR proteins, yet certain splicing factors can be mostly nucleoplasmic. To examine whether the distribution of various splicing factors in the nucleoplasm or nuclear speckles is modified upon lytic reactivation of KSHV, and to verify whether some of those factors localized to the enlarged SRSF2 speckles, we stained cells with antibodies to factors that participate in the machinery of gene expression. We examined two proteins that are known components of nuclear speckles, SON and Prp8. SON has an SR containing motif, acts as a splicing co-factor and is essential for nuclear speckle structural integrity [48,49]. Prp8 is a core splicing protein that coordinates critical rearrangements at the catalytic core of the spliceosome [50], and is regularly found in nuclear speckles [33]. In iSLK BAC16 mCherry-ORF45-infected cells undergoing lytic viral replication, both SON and Prp8 arranged in an aggregated manner resembling the distribution seen for SRSF2 and forming only where marginalized chromatin and mCherry-ORF45 markers appeared (Appendix A). Co-staining with SRSF2 followed by Pearson co-localization analysis showed that both proteins maintained their localization in nuclear speckles also after lytic infection (SRSF2:SON Pearson correlation coefficient r = 0.95; SRSF2:Prp8 r = 0.88) (Figure 2A–C).

Next, we stained the cells for several splicing factor components related to the U snRNP family: U1A, a core spliceosomal protein and a component of the U1 snRNP that is usually found in the nucleoplasm and in nuclear speckles [51,52]; SF3B1, which is a core component of the U2 snRNP, normally distributed in the nucleoplasm with some association to the periphery of nuclear speckles [51]; and U2AF65, which is a non-snRNP protein involved in the binding of U2 snRNP to the branch site of the intron, typically observed in the nucleoplasm and also in nuclear speckles in untreated cells [53]. U1A was occasionally found in the nuclear speckles (SRSF2:U1A r = 0.74), SF3B1 was also found co-localized to the foci (SRSF2:SF3B1 r = 0.81) during the lytic viral cycle (Figure 3A–C and Figure 2C).

It is important to note that splicing of viral RNAs continues during the lytic cycle of KSHV infection [54] and since it is known that transcription or splicing inhibition can affect nuclear speckle structure [18], we examined whether inhibition of splicing might have an effect on the aggregation of nuclear speckles. Indeed, the inhibition of splicing by Pladienolide B (PLB) caused the rounding up of nuclear speckles in untreated latently infected cells (Appendix A, top row), and no special phenotype was observed for the enlarged nuclear speckles during lytic reactivation (Appendix A, bottom row). This suggests either that the enlarged nuclear speckle structure observed in cells undergoing lytic infection is a unique structure formed under lytic viral reactivation or that PLB in infected cells also drives the formation of these enlarged nuclear speckles.

### 3.3. RNA Pol II Is Not Found in Nuclear Speckles During Lytic Reactivation of KSHV Infection

Although mRNA transcription most likely does not occur within the nuclear speckles, a number of studies have shown that it is possible to identify subunits of RNA Pol II in these structures [48,55,56,57]. We stained cells for endogenous RNA Pol II with an antibody that identifies Serine 5 phosphorylation of the C-terminal domain (CTD) of the large subunit of the polymerase. This phosphorylation is an indicator of RNA polymerase II during transcription initiation. However, this form of the polymerase did not colocalize with the nuclear speckles in reactivated cells compared to control ones (Figure 4A). The exclusion of the initiating RNA Pol II from SRSF2-contianing nuclear bodies within the nuclear volume was further documented by 3D rendering of the nucleus (Figure 4B and Appendix A).

We then examined the distribution of other RNA binding proteins participating in RNA Pol II gene expression pathways to determine whether they may also form an irregular nuclear structure upon lytic reactivation. During splicing, the exon junction complex (EJC) is deposited on the pre-mRNA close to the ligated exon-exon junctions [58]. An EJC component called Y14 is found in the nucleoplasm and in nuclear speckles (Appendix A top row). However, upon induction of lytic infection, the Y14 signal almost completely disappeared from the nuclei of infected cells (Appendix A bottom rows). Western blotting for Y14 showed no significant change in protein levels after reactivation (Appendix A).

### 3.4. Changes in the Distribution of mRNA Export Factors During Lytic Reactivation of KSHV Infection

The TRanscription and EXport (TREX) complex that is important for mRNA export is also deposited on the mRNA during transcription and while splicing occurs [59,60,61]. During lytic reactivation, the TREX component Aly/REF that is usually distributed in the nucleoplasm (Appendix A top row) was distributed in the whole nucleus (Appendix A bottom rows). Another TREX component, UAP56, which is also usually seen in the nucleoplasm and in nuclear speckles (Appendix A top row), was distributed in the nucleus of cells undergoing lytic infection (Appendix A bottom rows). Finally, the mRNA must reach the nuclear pore complex (NPC) for export. We examined the distribution of two nucleoporins, Nup153 of the NPC basket and Nup62 of the inner NPC basket ring (Appendix A). Both proteins remained in the NPCs during lytic reactivation, yet some clustering of NPCs in the nuclear envelope could be observed (Appendix A). Clustering of NPCs during herpesvirus infection has been documented previously [62,63].

### 3.5. Inhibition of KSHV DNA Replication Did Not Impair Nuclear Speckle Reorganization Upon Lytic Induction

In order to determine whether the subnuclear reorganization of nuclear speckle structures observed during the lytic reactivation of KSHV infection depends on virus DNA replication, we selectively inhibited the viral DNA polymerase with Phosphonoacetic acid (PAA) [64,65], which also inhibits late viral gene expression. Staining for the small capsid protein ORF65, representing late gene products, revealed that this marker appears during lytic reactivation but is not expressed in PAA treated cells (Appendix A). Control and reactivated iSLK-infected cells showed the expected SRSF2 foci (Figure 5A,B), yet upon lytic induction in the presence of PAA, some cells presented the familiar pattern of enlarged SRSF2 foci (Figure 5C top row), whereas in some cases, where chromatin marginalizing was observed, the nuclear speckles were not enlarged (Figure 5C bottom row). This could be due to the heterogeneity in the stage of lytic reactivation, as described above. The fact that SRSF2 enlarged nuclear speckles appeared even while inhibiting the viral late phase suggests that intra-nuclear reorganization during the lytic stage can occur independently from viral DNA replication and expression of late genes.

### 3.6. Poly(A)+ RNA Nuclear Distribution Changes during Lytic Reactivation of KSHV Infection

Nuclear speckles contain poly(A)+ RNA, yet the nature of these RNAs is not fully understood. These transcripts are probably not all mRNAs, and some are rather lncRNAs [30]. Using RNA FISH with an oligo-dT probe that hybridizes with the poly(A) tail of RNAs, we observed the expected high colocalization between the oligo-dT and SRSF2 in nuclear speckles of untreated cells (Figure 6A,E). In cells induced to the lytic cycle, we found that nuclear speckles could still be found with the poly(A) RNA signal (Figure 6D); however, in some nuclei, the two signals had completely separated from each other into two distinct structures (Figure 6B,C) and colocalization decreased to ~40% (Figure 6E).

The poly(A) tails of RNA are typically bound by the nuclear form of the poly(A)-binding protein PABPN1, a protein that is also observed in nuclear speckles together with poly(A)+ RNA [66]. We first examined its localization coefficient relative to nuclear speckles and found a slight decrease upon lytic reactivation from r = 8.5 to 7.6 (Figure 7A–C and Figure 2C). Moreover, we detected a decrease in the number of PABPN1 foci from approximately 38 foci to ~10 per nucleus (Figure 7B,D). We then proceeded to examine the fate of PABPN1 relative to poly(A) foci and found that they mostly remained together during lytic reactivation (Figure 7E), namely, the poly(A)+ sub-structures that were separated from the nuclear speckles contained the PABPN1 protein as well.

Nuclear speckles are known to contain the lncRNA MALAT1 [67]. Paraspeckles are another type of nuclear body adjacent to nuclear speckles that contain the lncRNA Neat1 [68]. To determine whether these lncRNAs distribute to any of the foci identified above, we performed RNA FISH to MALAT1 and Neat1. After induction of lytic reactivation, MALAT1 was dispersed in the nucleoplasm while Neat1 was undetectable (Figure 8), probably due to its initial low detection levels in untreated cells.

### 3.7. The Localization of Viral mRNAs during Lytic Reactivation of KSHV Infection

Since the poly(A) signal in cells in the lytic cycle was detected either in nuclear speckles or adjacent to nuclear speckles, and did not harbor the two lncRNAs we tested, we examined whether the polyadenylated RNAs might be viral mRNAs. We designed probe sets for single molecule RNA FISH against three viral transcripts encoding ORF29, ORF57, and ORF50 that undergo splicing and have crucial functions during the viral lytic cycle. ORF29 is a late viral gene product involved in packaging of the viral genomic DNA into the capsid [69]. ORF57 is an early multifunctional viral gene product that regulates the lytic cycle at the post-transcriptional level [70,71,72]. ORF50 is an immediate-early lytic gene product that functions as that functions as a key regulator of replication and transcription during the lytic cycle. RNA FISH probes that hybridize with *orf29* and *orf57* mRNAs were designed to detect the spliced transcripts. BAC16-mCherry-ORF45-infected iSLK cells showed the mRNA signal only in cells undergoing lytic infection, identified through the expression of mCherry-ORF45 protein signal and the condensed peripheral nuclear chromatin (Appendix A). Transcripts were both nuclear and cytoplasmic as expected. Examination of mRNA nuclear distribution in conjunction with the SRSF2 nuclear speckles showed that occasionally some *orf29* and *orf57* spliced mRNAs were adjacent, or even localized to the periphery of the nuclear speckles (Figure 9A,B). We also examined the sites *orf50* mRNA viral transcription using probes to the intron sequence of *orf50*. Introns are mostly detected at sites of transcription [33,73,74]. Signals were detected throughout the nucleus (Appendix A), in many foci suggesting that viral transcription is occurring at multiple loci. Here, too, occasionally some of the signal was found at the periphery of nuclear speckles (Figure 9C).

## 4. Discussion

In this study we examined the localization of RBPs involved in the gene expression pathway in the nuclei of cells undergoing productive lytic KSHV infection that show major alterations in nuclear structure. Upon induction of the lytic cycle of KSHV infection, the virus must utilize an arsenal of the host’s transcription and splicing factors, as it relies mostly on these factors in order to produce mature viral mRNA molecules. For instance, a variety of cellular splicing and export factors have mutual regulation with KSHV ORF57 protein, involved in splicing and export of viral mRNAs [75,76,77]. In the mammalian nucleus under regular conditions, many of the RBPs involved in pre-mRNA splicing and mRNA nucleocytoplasmic transport pass through nuclear speckles. The function of these subnuclear structures is not completely clear but they are usually found in the more internal parts of the nucleus, and not at the nuclear periphery, and can be in close association with areas of active gene [27,31,78].

Upon examination of the distribution of several splicing factors in cells undergoing the lytic infection cycle of KSHV, we documented redistribution of certain factors throughout the nucleus. For instance, 48 h after lytic induction the number of nuclear speckles was dramatically reduced and displayed few unusually large structures assumed to be generated by the fusion of several nuclear speckles. The enlarged nuclear speckles appeared close to the condensed chromatin, which was usually found at the periphery of the nucleus but could also be more internal. Notably, under regular conditions, nuclear speckles are found mostly in the nuclear center and not in the nuclear periphery. Other studies of KSHV infected cells have found partial colocalization of the viral ORF57 protein with nuclear speckles [54,75]. A similar phenomenon of redistribution of some RBPs to the nuclear periphery and their clustering has been documented during the lytic phase of herpes simplex virus 1 (HSV-1) and 2 (HSV-2), although the dramatic aggregation into few large nuclear speckles has not been reported [79,80,81]. Rather, many nuclear speckles were shown to decorate the nuclear periphery, while viral proteins such as ICP4, ICP8, ICP5, and p40 were not found in these nuclear speckles. Interestingly, growing replication compartments (RCs) followed in living HSV-1-infected cells showed that RCs were found adjacent to nuclear speckles at already early time points, before nuclear speckles localized to the periphery [82]. It is possible that RC formation in the center of the nucleus pushes the nuclear speckles towards the nuclear periphery of the nucleus; however, we found that blocking the actual event of KSHV viral replication did not stop the formation of the enlarged nuclear speckles and the marginalization of chromatin. Nuclear speckles are usually not found in direct contact with DNA under regular conditions, and thus, their presence at condensed chromatin in infected cells might be due to the forces pushing them to the periphery, especially since RCs movement has been shown to be an active process dependent on nuclear actin and myosin [82]. The latter study suggested that the proximity between RCs and nuclear speckles may enhance the export of HSV-1 viral mRNA form the nucleus.

We examined several known nuclear speckles constituents (SRSF2, SON, Prp8, U1A, SF3B1, U2AF65, PABPN1) and most continued to localize in the enlarged nuclear speckles of cells undergoing lytic KSHV reactivation, especially SON and Prp8. Some of the enlarged nuclear speckles contained a significant polyA(+) RNA population as found under regular conditions, yet in ~56% of cases, the poly(A)+ RNA clusters were separated and adjacent to the nuclear speckles in distinct sub-structures, suggesting that the two structures had formed due to fission of the nuclear speckle compartment. The poly(A)-binding protein (PABPN1) that is bound to poly(A) tails accompanied the poly(A)+ RNA in 66% of the cases in the foci that were still present. We tried to determine the identity of this poly(A)+ RNA population; a question that remains mostly unanswered even under regular conditions when the poly(A)+ RNA population is known to localize to nuclear speckles. The polyA(+) RNA signal in these foci did not originate from the nuclear speckle lncRNA MALAT1 or from the paraspeckle lncRNA Neat1, as they were dispersed throughout the nucleoplasm of the infected cells. MALAT1 has been previously shown to disperse in the nucleoplasm in response to different stresses [83,84], and nuclear speckles continued to form even when MALAT1 was lacking from the nuclear bodies [85], as observed here as well. The viral *orf29* and *orf57* spliced mRNAs dispersed throughout the cell and occasionally could be seen in the peripheral layer of nuclear speckles and seldom even more internally. Super-resolution microscopy has shown that the internal part of the nuclear speckle contains SRSF2 under regular conditions, while the peripheral layer normally contains MALAT1 and snRNAs [86]. Since MALAT1 is removed from the enlarged nuclear speckles of lytic cells, it is possible that this allows for the interaction of the nuclear speckle periphery with other RNAs such as viral mRNAs. Importantly, other studies that used RNA FISH to examine the transcription and distribution of other types of KSHV mRNAs in infected cells with respect to the viral RCs, found that they accumulated in foci around the RCs that colocalized with the RNA Pol II (serine 5 phosphorylated), and that the mRNAs required active DNA synthesis to accumulate [15,47]. A host mRNA that was tested did not accumulate in these nuclear foci, and a KSHV lncRNA (PAN) accumulated in foci differed from those with mRNA. Moreover, the mRNA nuclear foci were found in close proximity to SRSF2 nuclear speckles [47]. When we examined the transcription sites of *orf50* mRNA, they were found throughout the nucleus and in some cases could be seen in proximity to the nuclear speckles. Thus, it still remains unknown which other RNAs might be in these structures—cellular or viral RNAs. In any case, RNA Pol II also did not colocalize with the nuclear speckles, implying that these structures are not associated with transcription per se.

Why do nuclear speckles become round and enlarged during the lytic cycle of KSHV infection? It is known that the shutdown of transcription and subsequently pre-mRNA splicing in non-infected cells result in structural changes in the distribution of RNA binding proteins and nuclear speckles [87,88,89,90,91,92]. Namely, the usual irregular shape of the nuclear speckle becomes rounded and somewhat enlarged. This is thought to be the result of the inhibition of splicing such that splicing factors are no longer moving in and out of nuclear speckles, and instead are only returning to the nuclear speckles to be stored [93]. It is possible that since splicing factors are mostly not required during the lytic infection, since only a small fraction of KSHV gene repertoire contains introns, that some splicing factors tend to become stored in these large nuclear speckles whereas others disperse in the nucleoplasm. This abnormal segregation and redistribution of splicing factors would also imply that host gene expression is abrogated since not all splicing factors are available in the nucleoplasm. We have recently shown that nuclear speckles under normal conditions buffer the availability of splicing factors that are ready to engage in splicing in the nucleoplasm, and that this can affect splicing efficiency and the release of the nascent transcript from the gene after transcription is completed [24,33]. It is possible that the proximity of the enlarged nuclear speckles at the vicinity of the nucleus might still be useful for the few splicing events required for KSHV transcripts. It has been shown previously that intronless mRNAs can transit through nuclear speckles in order to obtain export competence [94], and thus, it is quite possible that viral RNAs use this pathway too during infection by utilizing the enlarged nuclear speckles for viral mRNA export [62]. Indeed, the examination of the distribution of NPC components did not detect any major difference in the cells transcribing viral RNAs, and thus, the pathway of viral mRNA export is probably not disrupted, though further assays in this matter are required.

## 5. Conclusions

In this study we examined the fate of nuclear speckles and RNA-binding proteins involved in pre-mRNA splicing and mRNA export during late stages of lytic KSHV infection. Lytic infection caused the segregation of proteins and RNAs from nuclear speckles and the redistribution of several factors in the nucleus, while certain splicing factors were retained in nuclear speckles that became few and enlarged nuclear bodies localizing to the periphery of the nucleus adjacent to condensed chromatin. Since only a small fraction of KSHV mRNAs undergo splicing, it seems that the virus preserves most of the host proteins so that it can utilize them; however, in parallel it causes segregation of the complexes such that they probably become less functional with relation to splicing but may have a role in the export of viral mRNAs from the nucleus due to their proximity to the nuclear periphery.

## Figures and Tables

**Figure 1 cells-09-01958-f001:**
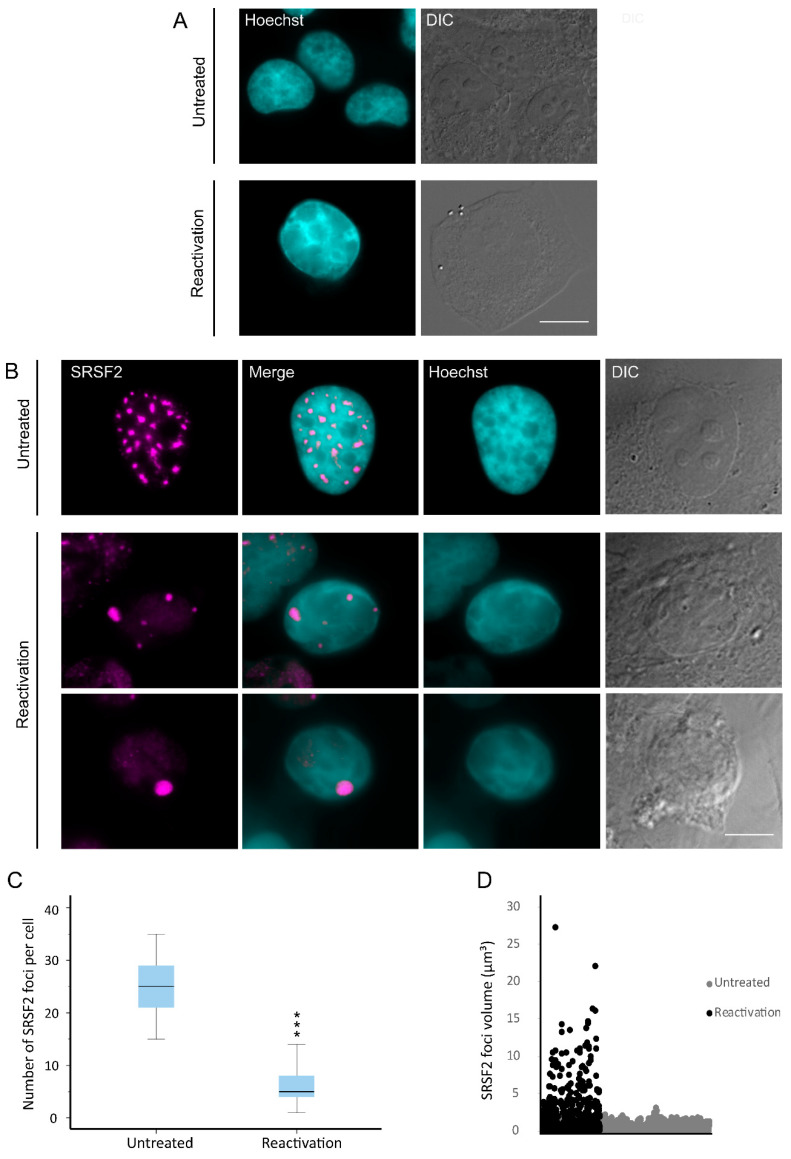
Redistribution of nuclear speckles to the nuclear periphery during lytic reactivation of KSHV. BAC16-infected iSLK cells were treated with Dox and n-Butyrate for 72 h to induce lytic reactivation. Untreated infected iSLK cells were used as a control. (**A**) Condensation and marginalization of chromatin was evident upon lytic reactivation (bottom row) by Hoechst staining (cyan; DIC in grey) as compared with control uninduced cells (top row). Nuclear speckles were stained with anti-SRSF2 (magenta). (**B**) A reduction in the number of nuclear speckles in conjunction with the appearance of enlarged foci in the nuclear periphery (second and third rows) was seen upon lytic reactivation, compared with untreated cells (top row). Bar, 10 µm (**C**) The numbers of nuclear speckles were counted and an independent samples t-test (two-tailed) was performed. For the box plot: center line, median; box limits, upper and lower quartiles; whiskers, minimum to maximum range. There was a significant difference between untreated latently-infected cells and cells undergoing lytic reactivation (*** *p* < 0.001) (nUntreated = 50, nReactivation = 50 cells). (**D**) The volumes of nuclear speckle foci were measured in three dimensions (3D), where a significant difference was again detected (nUntreated = 74, nReactivation = 41 cells) (*** *p* < 0.001).

**Figure 2 cells-09-01958-f002:**
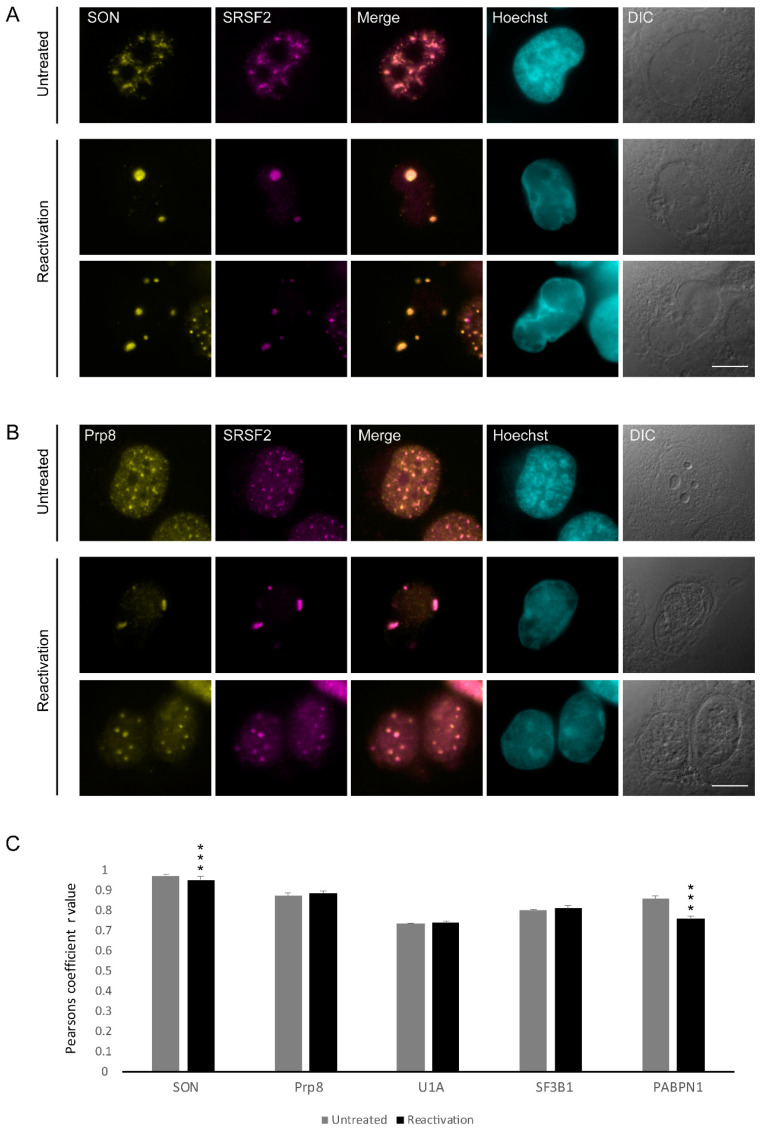
The distribution of SON and Prp8 during lytic reactivation of KSHV infection. BAC16 mCherry-ORF45-infected iSLK cells were treated to induce the lytic cycle of KSHV infection for 48 h and stained with (**A**) anti-SON or (**B**) Prp8 (yellow) together with SRSF2 (magenta); Hoechst in cyan; DIC in grey. Top row—untreated cells, bottom rows—reactivated cells. Bar, 10 µm. (**C**) Pearson’s colocalization coefficient for the comparison of colocalization between SRSF2 and: SON (nUntreated = 116, nReactivation = 107 cells), Prp8 (nUntreated = 71, nReactivation = 65 cells), U1A (nUntreated = 112, nReactivation = 83 cells), SF3B1 (nUntreated = 64, nReactivation = 59 cells), and PABPN1 (nUntreated = 99, nReactivation = 41 cells) (*** *p* < 0.001).

**Figure 3 cells-09-01958-f003:**
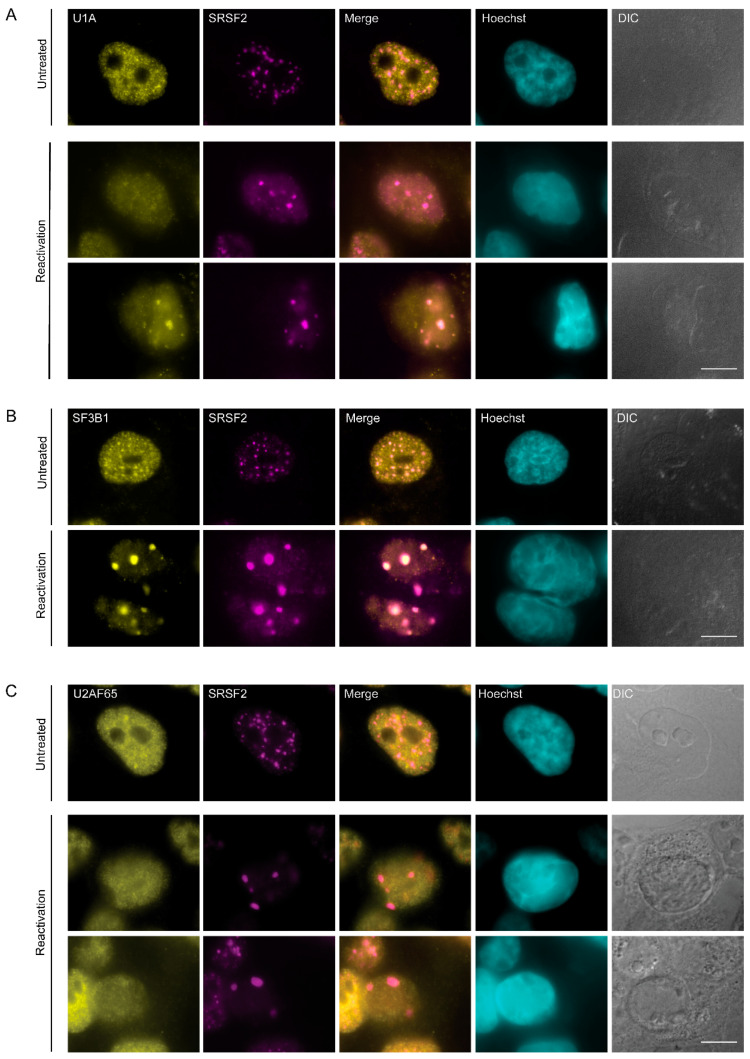
Distribution of splicing factors during lytic reactivation of KSHV infection. BAC16- infected iSLK cells were treated to induce the lytic cycle of KSHV infection for 48 h and stained with (**A**) U1A, (**B**) SF3B1, and (**C**) U2AF65 (yellow) together with anti-SRSF2 (magenta); Hoechst in cyan; DIC in grey. Top row—untreated cells, bottom rows—reactivated cells. Bar, 10 µm.

**Figure 4 cells-09-01958-f004:**
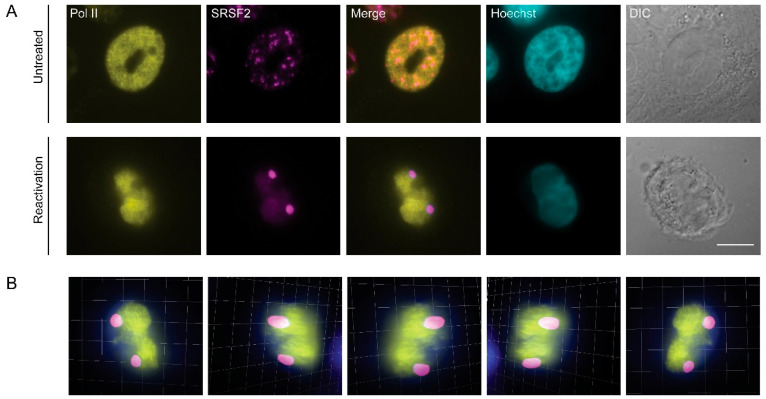
Initiating RNA Pol II is not found in nuclear speckles during lytic reactivation of KSHV. (**A**) BAC16-infected iSLK cells were treated to induce lytic reactivation for 48 h and stained with anti-RNA Pol II Serine 5 phosphorylated (initiating polymerase, yellow) and anti-SRSF2 (magenta); Hoechst in cyan; DIC in grey. Top row—control untreated cells. Bar, 10 µm. (**B**) Frames from Appendix A showing a 3D rendering of the volume of the cell in A.

**Figure 5 cells-09-01958-f005:**
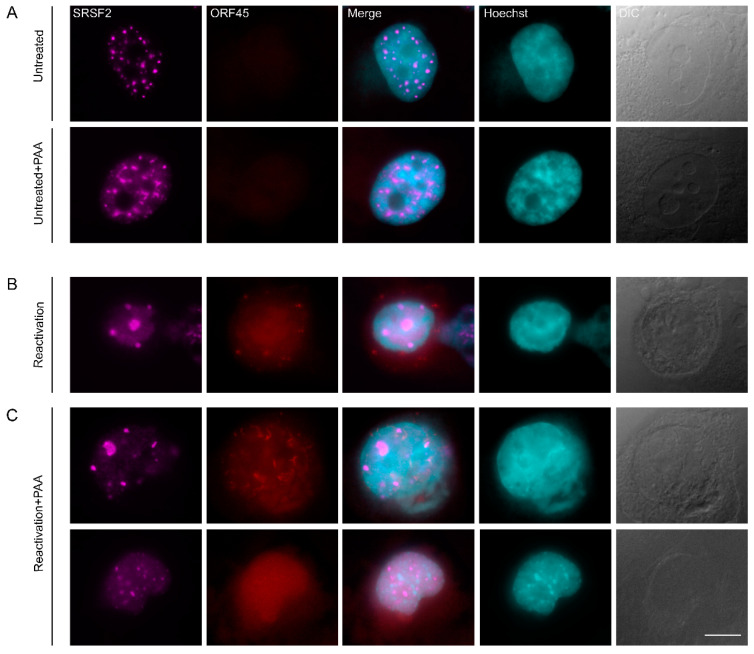
Inhibition of viral DNA replication and late gene expression does not prevent changes in nuclear structure. (**A**) Nuclear speckles (SRSF2, magenta) in BAC16-infected mCherryORF45 iSLK cells that were treated with PAA for 48 h, do not change their structure as compared to control untreated cells, whereas (**B**) induction of lytic reactivation for 48 h (detected by the expression of mCherry-ORF45, red) causes the formation of large nuclear speckles and chromatin condensation (Hoescht, cyan). (**C**) Treatment with PAA 2 h prior to lytic induction and during lytic reactivation does not prevent the formation of large nuclear speckles. Bar, 10 µm.

**Figure 6 cells-09-01958-f006:**
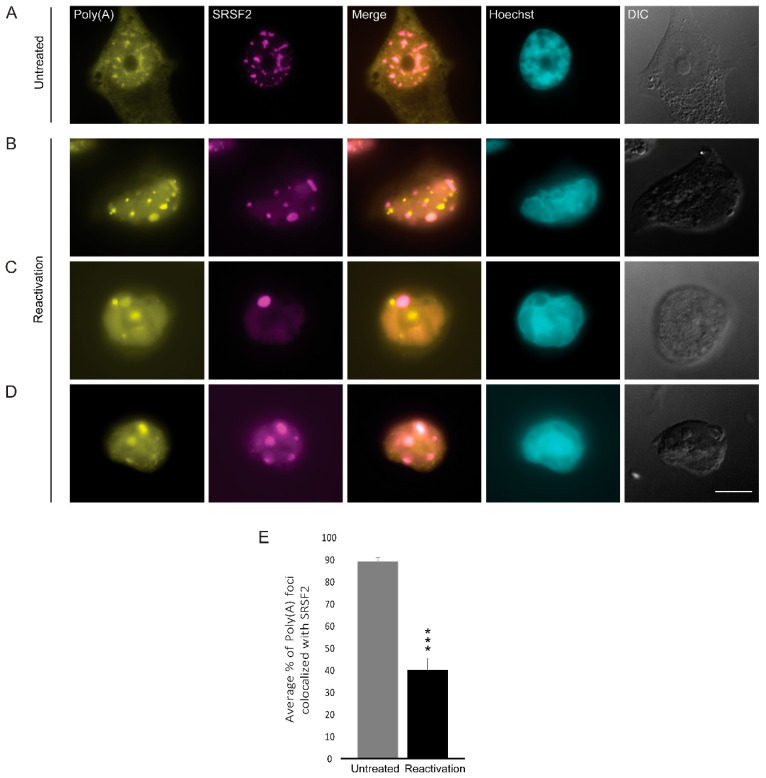
The distribution of poly(A)+ RNA during lytic reactivation of KSHV infection. Poly(A)+ RNA (yellow) was detected by RNA FISH using a Cy3-labeled oligo-dT probe (yellow) in (**A**) untreated cells compared to (**B**–**D**) BAC16-infected iSLK cells that were treated to induce the lytic cycle of KSHV infection for 48 h and stained with anti-SRSF2 (magenta); Hoechst in cyan; DIC in grey. Bar, 10 µm. (**E**) Measurements of the average percent of Poly(A) foci that colocalized with SRSF2 nuclear speckles (nUntreated = 55, nReactivation = 35 cells) (*** *p* < 0.001).

**Figure 7 cells-09-01958-f007:**
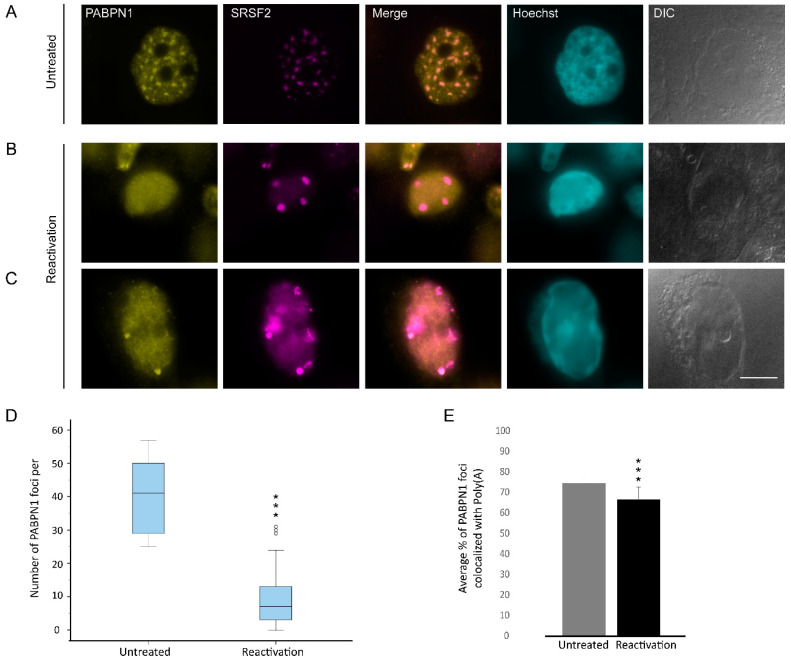
The distribution of PABPN1 during lytic reactivation of KSHV infection. PABPN1 (yellow) in (**A**) untreated cells compared to (**B**,**C**) BAC16-infected iSLK cells that were treated to induce the lytic cycle of KSHV infection for 48 h and stained with anti-SRSF2 (magenta); Hoechst in cyan; DIC in grey. Bar, 10 µm. (**D**) Measurements of the average numbers of PABPN1 foci per nucleus. For the box plot: center line, median; box limits, upper and lower quartiles; whiskers, minimum to maximum range. (*** *p* < 0.001) (**E**) The average percent of the PABPN1 foci that colocalized with the poly(A) foci (nUntreated = 75, nReactivation = 34 cells) (*** *p* < 0.001).

**Figure 8 cells-09-01958-f008:**
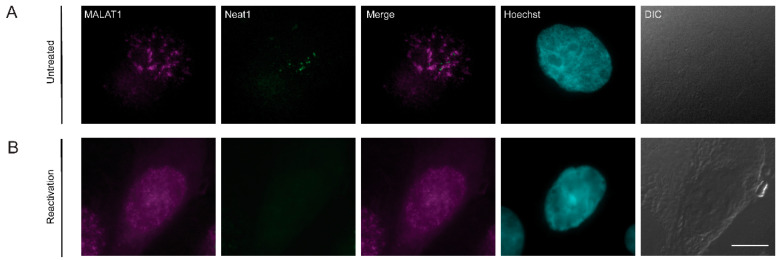
The distribution of lncRNAs during lytic reactivation of KSHV infection. MALAT1 (magenta) and Neat1 (green) lncRNAs were detected by RNA FISH in (**A**) untreated cells and (**B**) BAC16-infected iSLK cells that were treated to induce the lytic cycle of KSHV infection for 48 h. Hoechst in cyan; DIC in grey. Bar, 10 µm.

**Figure 9 cells-09-01958-f009:**
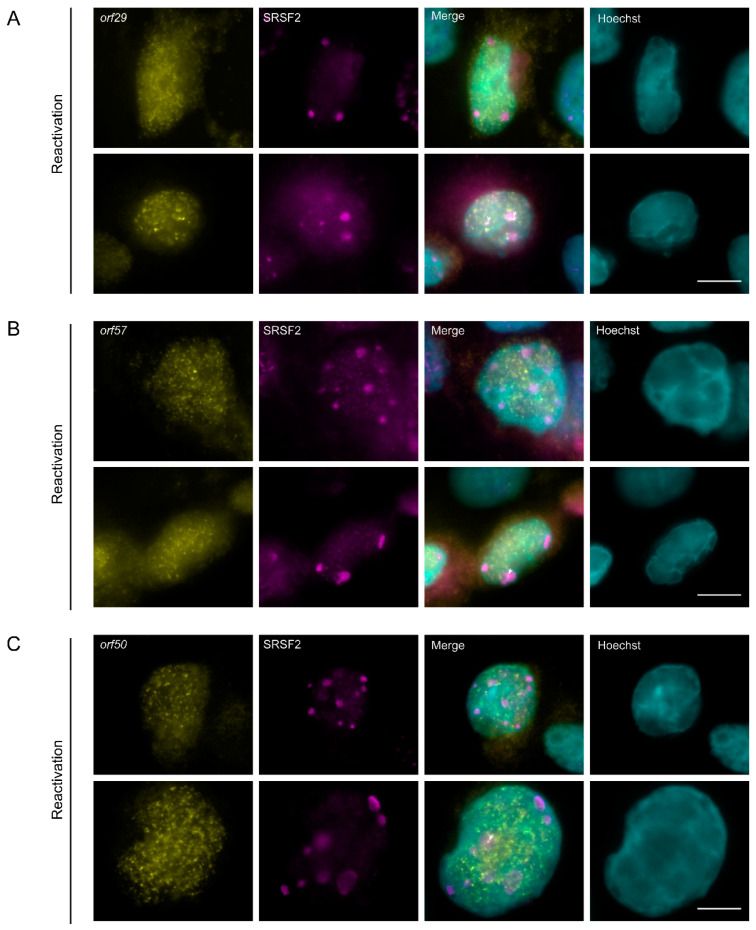
The distribution of spliced viral mRNAs during lytic reactivation of KSHV infection. (**A**) *orf29* and (**B**) *orf57* mRNAs were detected by RNA FISH (yellow) in BAC16 infected iSLK cells that were treated to induce the lytic cycle of KSHV infection for 48 h. (**C**) *orf50* (yellow) sites of transcription detected by RNA FISH to the intron of *orf50* pre-mRNA. SRSF2 in magenta; Hoechst in cyan. Arrowheads point to nuclear speckles that have viral RNA signal within them. Bar, 10 µm.

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
