# Peer review of "The Sub-Nuclear Localization of RNA-Binding Proteins in KSHV-Infected Cells"

_cells, 2020, doi:10.3390/cells9091958_

Round 1

Reviewer 1 Report

I am quite satisfied with the revision.

Reviewer 2 Report

The authors addressed my concerns in their revision, and while it is a descriptive study, I feel this is publishable in Cells as a foundation for future mechanistic studies. 

This manuscript is a resubmission of an earlier submission. The following is a list of the peer review reports and author responses from that submission.

Round 1

Reviewer 1 Report

Kaposi’s sarcoma-associated herpesvirus (KSHV) is a member of the lymphotropic gamma herpesvirus subfamily and one of the known oncogenic human viruses. KSHV can infect a wide variety of human cell types such as keratinocytes, epithelial, endothelial and lymphoid cells. Upon viral infections, as the nucleus fills up with compartments of virus replication, there are major changes in the distribution of the host cell chromatin, RNA and proteins. In this study, the authors determined the subcellular localization of a number of RNA-binding proteins and nuclear speckles during the reactivation of KSHV lytic cycle. The authors observed major changes in the numbers, sizes and subnuclear localization of nuclear speckles, splicing factors, polyadenylated RNA and lncRNAs.

There are three major concerns of this manuscript.

First, in all the experiments, BAC16 -infected iSLK cells were treated with Dox and Butyrate to induce lytic reactivation. And untreated infected iSLK cells were used as a control. After that, the cells were stained by antibodies or fluorescent nucleic acid probes. However, a critical control is the uninfected iSLK cells treated with Dox and n-Butyrate. Otherwise, it is unknown whether the observed changes are caused by viral reactivation or something else.

Second, the authors examined many factors for their subcellular localizations following the chemical treatment. However, there is no mechanistic or causative investigation of the changes, other than superficial and descriptive observations. It is unclear which one is the direct downstream effect of viral reactivation. Or which part or component of the viral life cycle play roles to elicit these changes.

Third, vice versa, it is unclear if any of these observed changes have functions or regulatory roles in viral life cycle, which actually is also very informative.

Minor points:

The numbers and sizes of foci or speckles should be quantified in all experiments.

Figure 2B and 2C can be combined into one box plot.

The format and typo should be carefully checked. For example, Line106, “5x105cells were grown on 18 cm coverslips”.

Reviewer 2 Report

The manuscript by Alkalay et al. addresses the important question of how the nuclear compartment and associated functions are reorganized upon activation of the KSHV lytic cycle. This is done using immunofluorescence and FISH to detect protein and RNA localizations within nuclei of KSHV active cells. The data presented indicates a number of changes to SRSF2 positive nuclear speckles involving shape, size, and composition. These changes are interesting and likely indicative of important biology going on within these cells. The reported findings are largely descriptive, and some are negative results, but do provide a basis upon which to develop more mechanistic questions and focused hypotheses. While I support publication of such data, there are a number of issues that I would like to see addressed through revision.

  1. A major issue is that most of the imaging data discussed in the text with respect to the main figures and supplements are supported by images that show one or just a few cells with no accompanying quantitation. This leads to much of the data being described with measures such as “some”, “numerous”, and “usually”. I strongly suggest some type of quantitation to address localization patterns and co-localization between tested factors to provide a more quantitative measure of the reported data.

  1. Another issue is around the use of a 48 hour time point. It is not discussed or shown what 48 hours is with respect to the viral lifecycle. At this time point, it could be that the viral program has switched to packaging and the vast majority of transcription/splicing is done. In a similar vein, are these observations reflective of the cell shutting down at this time point…being near death? Some discussion of these points would frame that data better and how it should be interpreted.

  1. On line 2776 the authors state, “ Indeed, the inhibition of splicing by Pladienolide B (PLB) caused the rounding up of nuclear speckles in untreated cells (Figure S1, top row), but did not change the phenotype of nuclear speckles during infection (Figure S1, bottom row). Therefore, the enlarged nuclear speckles observed in cells undergoing lytic infection is unique to viral infection.”. I do not agree this is the only interpretation. If the speckles are already in a molecular state that is like the state caused by PLB, i.e. no ongoing splicing, then the failure of these speckles to change does not indicate that they are unique. Rather, it indicates that PLB causes speckles in the treated cells to become KSHV-like speckles.

  1. With respect to Y14, the authors note the protein becomes undetectable, but this is not discussed anywhere else in the paper. Is it degraded? Is it modified and the Ab cannot recognize it? What about a western blot to address levels? At a minimum these types of questions should be raised in the discussion section.

  1. In Figure S4B, the nuclei shown as examples have abnormal shapes, could it be that there has been nuclear rupture? As such, are all nuclear components in the cytoplasm in these cells? What is the percentage of cells with a cytoplasmic signal in lytic cells?

  1. Figure 1/2 and 4/5 address the same questions and types of data, it may be useful to combine each pair of figures into a larger figure.

  1. The plot in figure 2B should show all individual data points. The table in 2C is not necessary and suggest that these numbers go on the plot or in the text.

Reviewer 3 Report

Authors observed localization of nuclear-speckles and several speckles-related factors. Unfortunately, there is not mechanistic insight, in other words, they just showed us localization information. It is hard to understand how nuclear-speckles change and what is going on inside the speckles upon infection.